# SM-TCN: Multi-Resolution Sparse Convolution Network for Efficient High-Dimensional Time Series Forecast

**DOI:** 10.3390/s25196013

**Published:** 2025-09-30

**Authors:** Ziyou Guo, Yan Sun, Tieru Wu

**Affiliations:** 1School of Artificial Intelligence, Jilin University, Changchun 130012, China; guozy22@mails.jlu.edu.cn; 2H. Milton Stewart School of Industrial and Systems Engineering, Georgia Institute of Technology, Atlanta, GA 30332, USA; yansun@gatech.edu

**Keywords:** high-dimensional time series forecasting, temporal convolutional network, sparsing models

## Abstract

High-dimensional time series data forecasting has been a popular problem in recent years, with ubiquitous applications in both scientific and business fields. Modern datasets may incorporate thousands of correlated time series that evolve together, and correctly identifying the correlated patterns and modeling the inter-series relationship can significantly promote forecast accuracy. However, most statistical methods are inadequate for handling complicated time series due to violation of model assumptions, and most recent deep learning approaches in the literature are either univariate (not fully utilizing inter-series information) or computationally expensive. This paper present SM-TCN, a Sparse Multi-scale Temporal Convolutional Network, utilizing a forward–backward residual architecture with sparse TCN kernels of different lengths to extract multi-resolution characteristics, which sufficiently reduces computational complexity specifically for high-dimensional problems. Extensive experiments on real-world datasets have demonstrated that SM-TCN outperforms state-of-the-art approaches by 10% in MAE and MAPE, and has the additional advantage of high computation efficiency.

## 1. Introduction

High-dimensional time series forecasting has been an important problem in many real-world applications, such as stock prediction [1,2], retail demand forecasting [3] and traffic forecasting [4], disease prediction [5,6], and robotics control [7,8]. An example is stock returns in S&P 500 index, while the return series has 500 dimensions, and most of the series could be explained by just a few factors. It is of considerable scientific and financial impact to provide accurate forecasting based on intra-series and inter-series correlation. However, accurate modeling of such in-context dependencies is often challenging for multivariate time series, due to complicated inter-series relationships when the dimension of the problem increases.

Traditional statistical methods operate on individual time series or a subset of time series, which are not scalable to high-dimension problems and can easily cause over-fitting issues. Some statistical methods for high-dimensional forecasting apply dimension reduction methods to select a subset of series with high correlation, such as LASSO regularization [9] and factor modeling [9] to transfer a high-dimensional problem to a regular forecast problem through a low-dimensional representation. However, statistical methods may struggle to model complicated nonlinear temporal correlation, as they rely on strong distribution assumptions.

Although deep neural network approaches have illustrated stronger ability of representation than statistical methods, they also have their respective limitations. Early works such as Recurrent Neural Networks (RNNs) and Long Short-Term Memory (LSTM) [10] networks may suffer from the gradient vanishing and exploding problems in training and are not readily available for multivariate or high-dimensional time series. Double residual networks have illustrated strong predictive power and achieved state-of-the-art performance in various forecast tasks. Such networks efficiently capture multi-resolution characteristics of the series and provide interpretable stack output from a series decomposition perspective, but as a univariate approach it applies simple concatenation on multivariate series as the input, and therefore fails to effectively capture correlation information among series, especially when the input dimension is large. Transformer-based approaches have achieved roaring success in time series forecasting, especially in long-time forecast (LTSF) tasks. However, most Transformer-based approaches only provide univariate prediction, and the most recent research has demonstrated that Transformer-based approaches often fail to capture even very simple causal dependencies among series, as they often struggle to achieve disentanglement between series. This indicates a clear gap: existing approaches either lack the ability to efficiently capture inter-series dependencies at scale, or fail to provide a principled way to couple sparsity, multi-resolution, and decomposition within a unified backbone.

To overcome such limitations, this work proposes SM-TCN, a Sparse Multi-scale Temporal Convolutional Network, which explicitly integrates sparse cross-channel convolutions, multi-resolution dilation, and a light continuity regularization into a single temporal–convolutional backbone, allowing each mechanism to play a distinct role rather than being a simple aggregation. SM-TCN addresses the high-dimension challenge through a regularized temporal convolutional neural network, which also effectively captures inter-series relationships to further enhance forecast accuracy. Moreover, by utilizing a backward-forward residual network with linear smoothing mechanism, SM-TCN decomposes original series into meaningful stack outputs, such as trends and seasonalities.

## 2. Summary of Contributions

In high-dimensional time series analysis, it is desirable for an approach to achieve the following properties:
(1)Computationally efficient and applicable for most forecast scenarios, especially when there is limited computation resources.(2)Effectively capture the inter-series relationship, while focusing on multi-scale series characteristics.(3)As a prerequisite to explore interpretability, the network should be extendable towards making its outputs human-interpretable, for example, be able to decompose series into interpretable partial outputs.

SM-TCN is, to our best knowledge, the first paper that accomplishes all the advantages above, thanks to the following highlights in the structure design.

### 2.1. Backward–Forward Output with Double Residual Stacking

We introduce a hierarchical framework to capture multi-resolution patterns from time series, especially from a long sequence history. The double residual stacking design also enhances the model interpretability from an series decomposition perspective [11], but the level of decomposition is automatically optimized without the necessity to pre-define the output basis functions.

### 2.2. Sparse Temporal Convolutional Neural Network for Multi-Resolution Extraction

We propose a sparse temporal convolutional neural network as an acceleration and regularization tool, which is shown to be effective in multivariate, especially high-dimensional problems. A sparse convolution operation effectively captures in-context information between series without incurring too much calculation cost. In each stack, this work chooses convolutional kernels with different sizes to capture multi-resolution inter-series patterns separately.

### 2.3. Linear Expansion Smoothing

We design a linear smoothing mechanism to project the convolution output to stack output, while only preserving the trend/seasonality characteristics. Through the introduction of continuity loss, this work harnesses the level of smoothness of different stack outputs to achieve a meaningful decomposition. A linear layer is also known to be cost efficient and easy to optimize.

## 3. Related Works

We mainly focus on the deep learning literature for time series analysis in this paper, and refer to [12,13,14,15,16,17,18] as statistical methods for time series. Based on application scenarios, time series techniques can be divided into three major categories: univariate techniques [2,19], multivariate techniques [20,21,22,23,24,25,26,27], and high-dimensional time series approaches [28,29,30]. Although univariate techniques can be applied to multivariate scenarios with simple series concatenation, they focus on each individual time series separately without extracting the correlations among time series, which limits the model’s performance. For example, FC-LSTM [31] forecasts univariate time series based on LSTM structure [10]. N-BEATS [11] proposes a deep backward–forward architecture based on a double residual network and fully connected layers with basis expansion. N-HITS [32] combines N-BEATS with a hierarchical interpolation technique to efficiently approximate arbitrarily long horizons.

Multivariate techniques consider a collection of multiple time series as a unified entity [9,25]. However, they are not easily scalable to large datasets, therefore may struggle when problem size is large due to high cost, over-fitting, and training difficulty. Temporal convolutional network (TCN) [20] utilizes 1D dilated convolution and treats the data entirely as a tensor input. LSTNet [33] combines convolution neural network (CNN) and recurrent neural network (RNN) to extract short-term local dependence patterns among variables and discover long-term patterns of time series. DeepState [23] combines parameterized linear state space models with a recurrent neural network (RNN). The introduction of Transformer [34] has enlightened many following studies in time series analysis, especially in long-horizon forecasting [35], which has dominated the landscape in the recent years. Most of the Transformer-based approaches (Autoformer [36], LogTrans [37], Reformer [38], Informer [39], FedFormer [40]) focus on improving computation efficiency and memory usage to improve prediction speed in long-horizon forecasting. Although Transformer-based approaches achieve dominant success in long-sequence forecasting, they still fail to dominate in the classical time series forecast tasks. Meanwhile, the newest literature [41,42] has brought the effect of Transformer architecture in time series forecasting to controversy. Recent work on Spatial Deep Convolutional Neural Networks (SD-CNN) [43] further emphasizes the role of spatial structures in time series modeling. Time series forecasting shares methodological relevance with structural detection tasks [44,45].

Modeling high-dimensional time series is always challenging. In particular, unregularized methods often suffer from the challenges of over-parameterization and lack of identifiability [46]. Consequently, deep learning approaches are often combined with dimension reduction or structural specification techniques. For example, ref. [28] is based on matrix factorization and is able to capture global patterns by representing each time series as a linear combination of basis components. The leading spectrum of the transition operator [47] is widely used to extract information from high-dimensional data with slow dynamics, such as a coarse-grained Markov state model. Another popular tool for dimension reduction is diffusion maps [48], which considers constructing a random walk on a graph and uses a spectrum of transition probability functions to define diffusion distance and coordinates for clustering and graph partitioning. Diffusion maps [49] can also be extended further to high-dimensional time series driven by stochastic differential equations for dimension reduction. Other techniques for high-dimensional analysis include the low-rank transition kernel representation method [50], LASSO regularization of VAR models [9], factor modeling [51], etc.

## 4. Methodology

An overview of the main structure of our framework is shown in Figure 1. For notation facility, our framework uses capital letters with bold faces *A* to denote matrix, bold faces *a* for vectors and regular letter A,a for real numbers. In many scientific scenarios, practitioners are more interested in the forecast of certain time series while treating other series as auxiliary data. Therefore, without loss of generality, our framework first divides the multivariate time series input as $I_{raw}=\left[X,A\right]\in R^{D_{raw}\times L}$, where $X={\left[x_{1:L}^{1},\ldots ,x_{1:L}^{N}\right]}^{\prime }\in R^{N\times L}$ denotes the subset of series (channels) for forecast and *L* is the historical length. $A={\left[a_{1:L}^{1},\ldots ,a_{1:L}^{D_{raw}-N}\right]}^{\prime }\in R^{(D_{raw}-N)\times L}$ denotes the series treated as exogenous variables (auxiliary data), where $D_{raw}=N+D_{A}$ is the raw input dimension (total number of channels). Our framework assumes the goal is to provide *H*-step-ahead forecast for series in *X*, denoted as $Y={\left[x_{L+1:L+H}^{1},\ldots ,x_{L+1:L+H}^{N}\right]}^{\prime }\in R^{N\times H}$, where *H* denotes the forecast horizon. In high-dimensional time series tasks, $D_{raw}$ can be very large, while in long time series forecast (LSTF), one may have H≫L. To overcome the limitations of existing methods that are incapable of efficiently dealing with multivariate, especially high-dimensional data, we propose an in-context multi-resolution framework, which incorporates channel pruning, double residual framework, a sparse temporal convolutional network, and linear expansion smoothing.

### 4.1. Channel Pruning

In high-dimensional analysis, over-representing inter-series dependencies may result in over-fitting as too much inter-series information is involved, especially when in high-dimensional datasets where inter-series correlation is weak. This paper proposes a simple yet powerful mechanism, *channel pruning*, which enforces the model to adaptively choose only the most significantly correlated time series, thus improving model stability and performance without additional computation cost. In this paper we adopt a hybrid channel pruning technology, which consists of two techniques: random dropping (hard dropping) and channel-wise attention mechanism (soft dropping) as shown in Figure 2.

#### 4.1.1. Random Dropping

We adopt a random dropping strategy to train multivariate time series without increasing computational complexity. In each ensemble sub-model, the original input $I_{raw}$, which contains $D_{raw}$ series, will be randomly shuffled and selected on channels of auxiliary data *A* to obtain pruned input $I=\left[\begin{array}{c} X\\ A^{1} \end{array}\right]$, where $A^{1}$ is the randomly selected channel into the first stack. The method only keeps $D_{prune}$ channels in total after pruning, where $D_{prune}<D_{raw}$. For notation facility, the method is denoted as $D≜D_{prune}$ in the following sections. From an ensemble perspective, in each sub-model, SM-TCN learns the contributions of a specific subset to the forecasting of series of interest. This random selection approach helps the model to identify the most influential series in the forecasting of other series, while efficiently alleviating the over-fitting issue. Combined with model ensemble, the method construct a pool of forecasting models to capture inter-series relationships at a subset level.

#### 4.1.2. Channel-Wise Attention Mechanism

Channel attention mechanism was first proposed in convolutional neural networks for exploiting the inter-channel relationship of features, and is widely used in computer vision [52,53,54]. This method aggregates the input channels and calculates an attention score for each input channel. Soft attention scores are adopted to alleviate the challenges in gradient propagation. For the stack input $I\in R^{D\times L}$, each input channel is first aggregated through a linear layer to obtain the corresponding vector $\gamma =W_{1}^{T}I+b_{1}\in R^{D}$, which then passes through a one-layer MLP with a modified sigmoid activation function to obtain a soft attention score for each channel *i*, which can be formulated as:
(1)$$\begin{array}{c} \gamma =\mathrm{Sigmoid}(W_{2}\cdot \mathrm{GELU}\left(W_{1}^{T}\cdot I+b_{1}\right)-\infty \cdot e+b_{2})+e \end{array}$$
where $e=\left(\underset{N}{\underset{︸}{1,1,\ldots ,1}},0,\ldots ,0\right)\in R^{D}$. Finally, the attention-enhanced input is obtained as $I_{att}=\gamma \cdot I$. Compared to conventional attention scores, one key advantage of Equation (Equation 1) is that the channel input of *X* will not be changed after multiplication as the score will always be 1, therefore preserving the properties of the residual network. Another advantage is that the information of all channels in *X* is preserved after the attention manipulation.

### 4.2. Sparse Multi-Scale TCN

The structure of the building components of each stack is introduced in Figure 1.

To illustrate, the first stack is taken as a special case. The input $I^{\prime }\in R^{D\times L}$ is reshaped to $R^{1\times L\times D}$, assuming that the input has *D* channels. Sparse kernels are then constructed in each stack *s* for 1D convolution with size $K^{s}\in R^{1\times W^{s}\times D\times N}$, where $W^{s}$ is the preset kernel width in stack s,s=1,2…,S as shown in Figure 3. The convolution is assumed to be performed with no padding zeros and stride equal to 1. Therefore, the output feature matrix of a convolutional layer $O^{s}\in R^{1\times (L-W^{s}+1)\times N}=K*X$ is given by:
(2)$$\begin{array}{c} O^{s}\left(1,l,n\right)=\sum_{d=1}^{D}\sum_{w=1}^{W^{s}}K\left(1,w,d,n\right)X\left(1,l+w-1,d\right),\\ \mathrm{for}\;l=1,2,\ldots ,L-W^{s}+1 \end{array}$$

While the conventional convolution operation might be expensive, the calculation is accelerated using a faster sparse version based on multiplication of sparse matrices. The key idea is to approximate dense matrix *I* with a low-rank matrix *J* and the convolutional kernel *K* with a low-rank matrix *R*, such that O=R∗J.

Viewing convolution as a linear operator, the approximation error can be bounded as
(3)∥K∗X−R∗J∥≤∥(K−R)∗X∥+∥R∗(X−J)∥
which shows that the total error stems from both kernel pruning (K→R) and input reduction (X→J). This bound explains why accuracy remains stable under moderate sparsity, but deteriorates if pruning or rank reduction is too aggressive.
(4)$$\begin{array}{c} K\left(1,w,i,n\right)\approx \sum_{d=1}^{D}R\left(1,w,d,n\right)P\left(d,i\right) \end{array}$$
(5)$$\begin{array}{c} \;J\left(1,l,i\right)=\sum_{d=1}^{D}P\left(i,d\right)X\left(1,l,d\right) \end{array}$$

In the next procedure, for each input channel d=1,2,…,D, $R\left(\cdot ,\cdot ,d,\cdot \right)\in R^{1\times W^{s}\times N}$ is decomposed into the product of a matrix $S_{d}\in R^{R_{d}\times N}$ and $Q_{d}\in R^{1\times W\times R_{d}}$, where $R_{d}$ is a small number that denotes the number of bases:
(6)$$\begin{array}{c} \;R\left(1,w,d,n\right)\approx \sum_{r=1}^{R_{d}}S_{d}\left(k,n\right)Q_{d}\left(1,w,k\right) \end{array}$$
(7)$$\begin{array}{c} T_{d}\left(1,l,k\right)=\sum_{w=1}^{W^{s}}Q_{d}\left(1,w,k\right)J\left(1,l+w-1,d\right) \end{array}$$

Summing up all the results above, the final approximation of the output matrix *O* is obtained:
(8)$$\begin{array}{c} O\left(1,l,n\right)\approx \sum_{d=1}^{D}\sum_{r=1}^{R_{d}}S_{d}\left(k,n\right)T_{d}\left(1,l,r\right) \end{array}$$

There are many advantages to using sparse convolution in high-dimensional scenarios. First, the computational complexity has significantly reduced to $\left(\gamma DN+\sum_{d=1}^{D}R_{d}\right)W\left(L-W+1\right)+D^{2}L$ [55]. Second, similar to regularization approaches, the introduction of a sparse kernel effectively alleviates over-fitting issue for high-dimensional series. Moreover, it is observed that sparsity in the range of 30–50% maintains stable performance, while 60% shows mild degradation and 70% degrades more clearly; thus, sparsity is typically kept in the mid-range and increased only when efficiency is prioritized.

The convoluted output will then go through a dilated temporal convolutional network (TCN). To focus on analyzing the components of the stack input with different resolution scales, a dilated 1D-convolution layer with kernel size $k_{i}$ is introduced. Larger kernel size $k_{i}$ will filter out high frequency components (local patterns) and force the stack to focus on low frequency (global) patterns. This acts as a first-difference continuity regularizer, which attenuates high-frequency components more strongly and stabilizes learning under sparsity. Inclusion of a convolution layer in each stack also helps to reduce the size of inputs to each stack and limit the number of learnable parameters in the network. Moreover, the multi-resolution mechanism can be seen as a regularization approach, which alleviates the effect of over-fitting, while still keeping the original receptive field. In the SM-TCN, stacks with larger index *i* are designed to capture higher frequency patterns. Therefore, the sequence of kernel size $k_{1},\ldots ,k_{N}$ should be monotonically decreasing. The outputs after convolution layer are denoted as $I_{Conv}^{s}$ for each stack *s*, with size $L_{conv}^{s}$ depending on the choice of kernel size $k_{i}$ and dilation parameters.

### 4.3. Linear Expansion Smoothing

The output size is aligned by transforming the length of $L_{conv}^{s}$ to L+H, generating a stack output which consists of both partial backcast and forecast without introducing too much computational complexity. The simple idea is to introduce a one-layer linear model to generate backcast and forecast time series directly. Such an idea has been proven simple yet powerful in many forecast tasks [11,32,42], outperforming complex Transformer-based models by a large margin, especially when extracting temporal relations from long sequences. The feature $X_{conv}^{s}$ is first decomposed into a trend component and a seasonal component, which are then fitted with a one-layer linear model separately. Finally, the two components are summed to obtain the final stack output ${\hat{Y}}^{s}\in R^{N\times H}$ and ${\hat{X}}^{s}\in R^{N\times L}$.

To control the variability of the stack output, a continuity loss term $L_{\mathrm{cont}}$ is applied to harness the smoothness of ${\hat{T}}^{s}≜\mathrm{concat}{\left[{\hat{X}}^{s},{\hat{Y}}^{s}\right]}^{T}\in R^{N\times (L+H)}$:(9)$$\begin{array}{c} L_{\mathrm{cont}}^{s}=\sum_{n=1}^{N}\sum_{s=1}^{S}\sum_{t=2}^{L+H}\frac{\beta^{s}}{(L+H)S}{\left(\frac{{\hat{T}}_{t,n}-{\hat{T}}_{t-1,n}}{\sigma_{n}}\right)}^{2} \end{array}$$
where $\beta^{s}$ is a factor that adjusts the influence of the continuity loss in the overall loss function. $\sigma_{n}$ is the empirical standard deviation of the $n^{th}$ channel in *X* for n=1,2,…,N. By applying the continuity loss, temporal details and noise are smoothed out, while overall trends described by inter-series dependencies are emphasized, and the remaining intra-series details are left to higher-level stacks.

### 4.4. Doubly Residual Network

Doubly residual topology is widely applied for time series analysis due to its efficient structure and high forecast accuracy. The architecture of the double residual network comprises two residual branches, one running over the backcast prediction of each layer and the other running over the forecast branch of each layer. Double residual networks can also alleviate the challenge of gradient back-propagation and are easier to optimize. More importantly, the smoothing linear expansion ensures that each block outputs a meaningful partial forecast with a different level of variability, representing a hierarchical decomposition of the series ranging from low volatility to high volatility. All partial forecasts obtained at each stack are then summed up to provide the final forecast. Following [11,32], a double residual network structure is adopted, where the input of each stack is defined as the difference between the input to the previous stack and the corresponding stack backcast. As previously mentioned, in the special case of the first stack, the stack input is the original time series after random dropping and the attention mechanism, i.e., $I^{1}=\left[\begin{array}{c} X^{1}\\ A^{1} \end{array}\right],I_{att}^{1}=\gamma \cdot I^{1}$, with $X^{1}=X$. For all subsequent stacks, the previous stacks remove the portion of the signal that can be well approximated.
(10)$$\begin{array}{c} I^{s}=\left[\begin{array}{c} X^{s-1}-{\hat{X}}^{s-1}\\ A^{s} \end{array}\right],I_{att}^{s}=\gamma \cdot I^{s},s=1,2,\ldots ,S \end{array}$$
where $A^{s}$ is independently selected from *A* in different stacks. The method then conduct a sequential analysis of the time series residual as the stack input, generating a backcast series to fit the historical input and a forecast series to predict the future series. The final output of the model is given by $\hat{Y}=\sum_{s=1}^{S}{\hat{Y}}^{s}$.

This work optimize the MSE loss between the ground truth $\hat{Y}$ and *Y*, while adding regularization to ensure the sparsity introduced through the kernel and smoothness (continuity) of the stack output. Our framework finally qcquire the final constrained optimization problem as:
(11)$$\begin{array}{cc} L= & \underset{\mathrm{data loss}}{\underset{︸}{{\left(\hat{Y}-Y\right)}^{2}}}+\underset{\mathrm{sparse decomposition loss, Section 4.2}}{\underset{︸}{\lambda_{1}\sum_{d=1}^{D}∥S_{d}{∥}_{1}+\lambda_{2}\sum_{d=1}^{D}\sum_{r=1}^{R_{d}}{∥S_{d}\left(r,\cdot \right)∥}_{2}}}\\ & +\underset{\mathrm{continuity loss}}{\underset{︸}{\mu \sum_{s=1}^{S}L_{\mathrm{cont}}^{s}}} \end{array}$$

## 5. Experimental Results

Comprehensive experiments were conducted on nine widely used multivariate time series forecasting datasets, including ETT, Traffic, Electricity, Weather, Exchange Rate, and Multi. This paper compared SM-TCN with six state-of-the-art baseline models: ARM, PatchTST, DLinear, FedFormer, Autoformer, and Informer. To ensure a fair comparison, the experimental setup was aligned with those used in these baseline models.

### 5.1. Training Methodology

In the SM-TCN configuration, the number of stacks is set to six, as experiments have shown this achieves the best balance between prediction accuracy and model complexity. The kernel width $W_{s}$ follows a moderate linear growth rate of $W_{s}=3,5,7,9,11$, ensuring a well-balanced receptive field. Sparse convolution is applied with a sparsity level of 30–50%, improving efficiency while maintaining performance. The dilation rate follows an increasing schedule, allowing the model to capture both short-term and long-term dependencies effectively. Preprocessing: per-channel z-score using training-split statistics only; missing values handled by forward fill + linear interpolation, with seasonal mean (hour-of-day/day-of-week) as fallback for long gaps; windows with unresolved entries are dropped.

SM-TCN is optimized for long-sequence forecasting, with the input tensor size set to a fixed lookback window of L=720, ensuring effective modeling of long-term dependencies without excessive computational overhead. The prediction window size *H* is adjusted based on task requirements, with a default of 96 for most time series forecasting applications. The batch size is set to 64, but for extremely long sequences (L>1000), the batch size is reduced to optimize memory usage. Channel pruning is implemented with a pruning ratio of 20%, which has been shown to improve stability and generalization. Splits and sampling: chronological train/validation/test; sliding window with lookback L=720, default horizon *H* = 96, training stride 1; windowing, batch size, and scheduler are identical across all baselines.

The model is trained using the AdamW optimizer with an initial learning rate of $1e^{-3}$, employing a cosine annealing scheduler over 100 epochs, with early stopping patience set at 10 epochs. The loss function consists of MSE loss with additional sparse decomposition and continuity regularization to ensure smoothness and structured feature selection. Gradient clipping is applied with a max norm of 5 to prevent exploding gradients. The model runs on a single NVIDIA RTX 4090 GPU, with a default dropout rate of 0.1 to prevent overfitting, while Xavier initialization and batch normalization are used as default settings.

Table 1 summarizes the forecasting performance of SM-TCN and several baselines across eight multivariate time series datasets under multiple prediction horizons. SM-TCN consistently achieves top results, ranking first in MSE on seven datasets and in MAE on six datasets. It attains the lowest average ranks (1.25 for both MSE and MAE) and outperforms all other models, especially on challenging datasets like ETTm1, Weather, and ETTh2. ARM is the closest competitor, ranking second overall and winning on the Exchange dataset.

Among Transformer-based models, iTransformer achieves relatively stronger results compared with FedFormer, Autoformer, and Informer, but still falls short of SM-TCN and ARM. PatchTST and DLinear obtain moderate results but fail to consistently surpass SM-TCN or ARM. Overall, the results demonstrate SM-TCN’s strong generalization ability and robustness across diverse forecasting tasks.

### 5.2. Ablation Study

In this subsection, we conduct ablation studies on the four primary modules of SM-TCN to demonstrate their specific mechanisms for performance enhancement. Experiments are conducted under the same settings, with the input length fixed at 720. Specifically, this work compares the full SM-TCN model with versions where each of the four primary modules is removed, and the experimental results across different datasets are shown in Table 2. Furthermore, we perform additional ablation studies on the internal details of each module, such as hyperparameters.

#### 5.2.1. Efficacy of Channel Pruning

Channel pruning significantly enhances forecasting accuracy and computational efficiency by selectively retaining the most relevant channels, as shown in Table 3. The pruned model demonstrates better generalization to unseen data, effectively reducing overfitting and improving overall performance compared to models without pruning. To achieve an optimal trade-off between accuracy and computational cost, the main model adopts a pruning ratio of 20%, ensuring both high efficiency and reliable predictions.

Further analysis in Table 3 reveals the impact of different pruning levels, indicating that a 20% pruning ratio provides the best balance. At this level, the model retains the most informative features, achieving the lowest prediction error and the highest computational efficiency. Lower pruning ratios preserve more information but introduce redundancy that weakens performance, while higher pruning ratios risk the loss of critical information, leading to decreased accuracy.

#### 5.2.2. Efficacy of Linear Expansion Smoothing

Linear expansion smoothing (LES) plays a crucial role in improving forecasting accuracy and stability by effectively capturing overall trends. The model with LES achieves lower prediction errors and more consistent results, while removing LES leads to increased errors and greater variability. These findings confirm the importance of LES in generating reliable forecasts, as shown in Table 2.

#### 5.2.3. Selection of Sparse Convolutional Kernels

Table 4 provides an analysis of how different growth rates of kernel width $W_{s}$ with respect to stack index *s* affect model performance. The table compares various linear growth rates and an exponential growth strategy.

The findings indicate that using a moderate linear growth rate, where $W_{s}$ increases by 2 units with each stack ($W_{s}=3,5,7,9,11$), results in the best prediction accuracy and model stability. Slower linear growth rates, such as increasing $W_{s}$ by one, are insufficient for capturing long-term dependencies, while faster rates, increasing by three or more, cause the model to miss short-term details.

In contrast, the exponential growth strategy ($W_{s}=3,6,12,24$) leads to excessive focus on long-term trends, neglecting critical short-term patterns, and thus performs the worst. This analysis suggests that a moderate, steady increase in kernel width is most effective for balancing short-term and long-term feature extraction.

#### 5.2.4. Number of Stacks

In the doubly residual network, the number of stacks plays a crucial role in balancing model complexity, prediction accuracy, and stability. Each stack in the network architecture progressively refines the input by removing components that can be well approximated, achieving a hierarchical decomposition from low to high volatility. By sequentially processing residual signals, the model gradually enhances the representation of underlying patterns, ultimately improving forecasting accuracy.

Table 5 presents the impact of different stack numbers (4, 5, 6, 7, and 8) on model performance. The results indicate that when the number of stacks is set to 6, the model achieves the optimal balance between accuracy and complexity. As the number of stacks increases from 4 to 6, the forecasting performance improves significantly, with a notable reduction in mean squared error (MSE). This improvement is attributed to the network’s enhanced ability to capture both short-term and long-term dependencies while effectively managing noise through the residual structure. However, beyond 6 stacks, the performance gains plateau, and further increasing the number of stacks leads to diminishing returns and potential overfitting, as excessive decomposition may capture noise rather than meaningful patterns.

### 5.3. Visualization

To gain insight into how the channel-wise attention evolves throughout training, we visualizes the attention weights assigned to each input variable at different training stages on the Weather dataset. As illustrated in Figure 4, this paper collect snapshots of the learned channel attention every 10 epochs, ranging from Epoch 10 to Epoch 100.

In the early stages (e.g., Epochs 10–40), the attention distribution is relatively dynamic, indicating that the model is still exploring the relevance of different input variables. As training progresses, the attention weights gradually stabilize and converge to a consistent pattern. From Epoch 80 onwards, the attention distributions remain largely unchanged, suggesting that the model has identified a reliable set of important variables. This observation aligns with our attention design in Equation (Equation 1), which softly reweights auxiliary channels while preserving the core forecasting input *X*. The visualization provides intuitive evidence that our model adaptively refines its focus on meaningful input features over time.

In addition to attention dynamics, we also visualize the sparsity pattern of the learned convolutional decomposition matrix $S_{d}$, which encodes the contribution of each input channel to the output channels in the sparse kernel structure. As shown in Figure 5, the kernel matrix exhibits a high degree of sparsity, with many weights pruned to zero. This confirms the effectiveness of the sparsity-inducing regularization terms in our loss function and highlights that the model learns to activate only a small subset of relevant input features, thereby improving interpretability and computational efficiency.

### 5.4. Computational Experimental

To comprehensively evaluate efficiency, this paper compares SM-TCN with representative Transformer-based baselines, including Autoformer, Informer, and iTransformer. This paper report parameters, FLOPs, average training time per epoch (s/epoch), and average inference time per batch (s/batch) under different horizons (H=160,320,480). Unless otherwise stated, all timings are measured on the Electricity dataset, on a single NVIDIA RTX 3090, with identical dataloaders and training loops. We average over three runs after warm-up and synchronize the GPU before timing; timings exclude I/O and validation to ensure fairness. Results are summarized in Table 6.

From Table 6, SM-TCN achieves a favorable trade-off between efficiency and size compared with Transformer-based baselines. Autoformer and Informer incur notably higher FLOPs and longer training/inference times, especially for larger *H*. iTransformer improves efficiency compared to other Transformer models, but still requires more FLOPs and time than SM-TCN. By contrast, SM-TCN maintains consistently lower computational cost while preserving accuracy, confirming its suitability for high-dimensional, long-horizon forecasting tasks.

## 6. Conclusions

In this work, this paper introduces SM-TCN, a Sparse Multi-scale Temporal Convolutional Network, to address the challenges of high-dimensional time series forecasting. Traditional statistical methods, while effective in low-dimensional settings, struggle with scalability and nonlinear dependencies, while deep learning approaches often fail to capture inter-series relationships effectively. To bridge this gap, SM-TCN combines a sparse temporal convolutional network and double residual stacking to enhance both efficiency and predictive accuracy.

SM-TCN leverages a hierarchical residual framework to extract multi-scale temporal patterns, allowing it to capture long-range dependencies while maintaining computational efficiency. Its sparse convolutional design reduces unnecessary computations, making the model scalable for high-dimensional forecasting tasks, while maintaining strong generalization capabilities.

Experimental results demonstrate that SM-TCN consistently outperforms existing methods in high-dimensional forecasting, providing efficient and accurate predictions. By balancing computational efficiency and inter-series correlation modeling, SM-TCN represents a significant advancement in time series forecasting. Future research will focus on further enhancing the model’s ability to decompose time series into interpretable trend components and disentangle causal relationships between series, paving the way for more transparent and robust forecasting models.

## Figures and Tables

**Figure 1 sensors-25-06013-f001:**
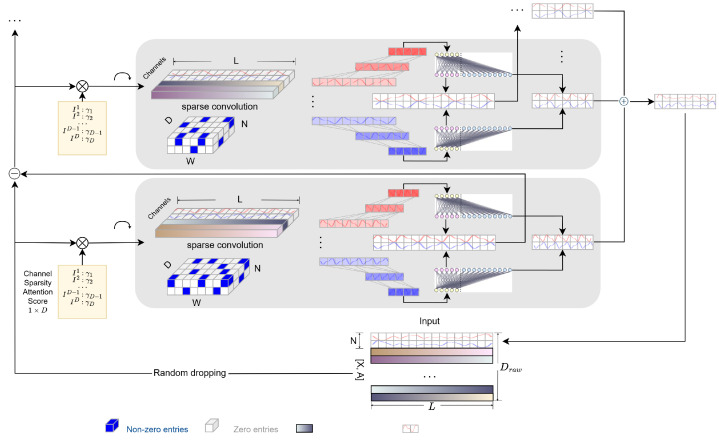
Overview of the proposed framework. The framework processes a multivariate time series input $I_{\mathrm{raw}}=\left[X,A\right]\in R^{D_{\mathrm{raw}}\times L}$, where $X\in R^{N\times L}$ represents the target series for forecasting, and $A\in R^{(D_{\mathrm{raw}}-N)\times L}$ denotes auxiliary data. The objective is to predict $Y\in R^{N\times H}$, an *H*-step-ahead forecast of *X*. Given the challenges of high-dimensional time series and long-term forecasting (H≫L).

**Figure 2 sensors-25-06013-f002:**
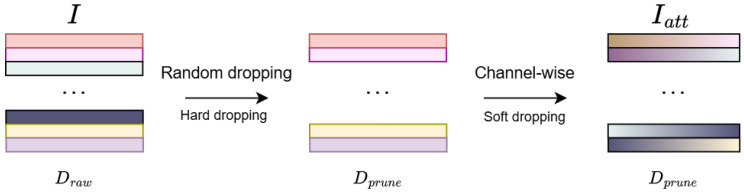
The input $I\in R^{D_{\mathrm{raw}}\times L}$ undergoes random dropping (hard pruning), where a subset of channels is randomly selected, reducing the dimension from $D_{\mathrm{raw}}$ to $D_{\mathrm{prune}}$. The pruned input $I\in R^{D_{\mathrm{prune}}\times L}$ is then processed by the channel-wise attention mechanism (soft pruning), which preserves the dimensionality. The final weighted input $I_{\mathrm{att}}\in R^{D_{\mathrm{prune}}\times L}$ maintains the same number of channels as *I*.

**Figure 3 sensors-25-06013-f003:**
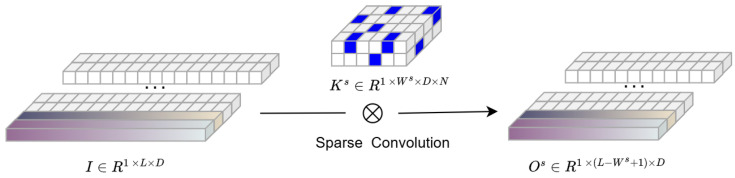
The input matrix $I\in R^{1\times L\times D}$ is transformed into the output matrix $O^{s}\in R^{1\times (L-W^{s}+1)\times N}$ through the sparse convolutional kernel $K^{s}\in R^{1\times W^{s}\times D\times N}$, which reduces computation while preserving key features.

**Figure 4 sensors-25-06013-f004:**
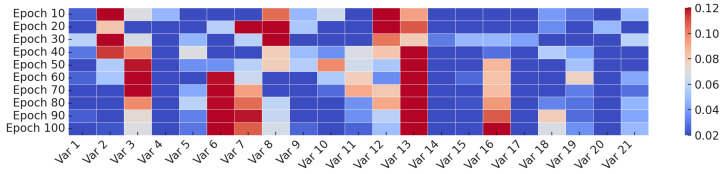
Evolution of channel-wise attention weights during training on the Weather dataset. Attention gradually shifts toward more informative variables.

**Figure 5 sensors-25-06013-f005:**
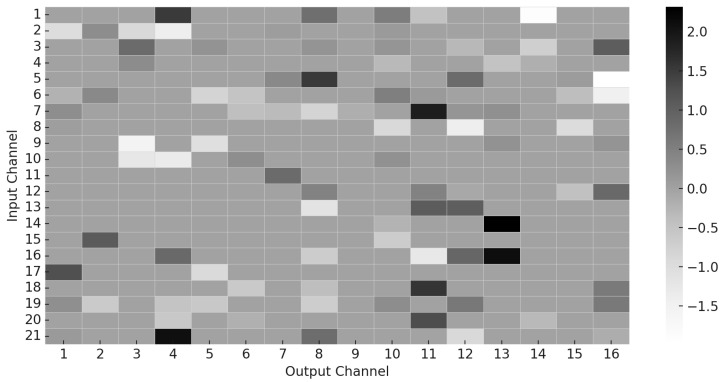
Sparsity pattern of the learned kernel matrix $S_{d}$. Only a subset of input channels contributes significantly to the output, validating the effectiveness of sparse regularization.

**Table 1 sensors-25-06013-t001:** A summary of multivariate time series forecasting results for SM-TCN and its comparison methods, including iTransformer, under prediction horizons L∈{96,192,336,720}. Average rankings (AvgRank) of each model and the count of first-place rankings (#Win) are also included.

Models	SM-TCN		ARM		iTransformer		PatchTST		DLinear		FedFormer		Autoformer		Informer	
Dataset	MSE	MAE	MSE	MAE	MSE	MAE	MSE	MAE	MSE	MAE	MSE	MAE	MSE	MAE	MSE	MAE
Electricity	0.149	0.244	0.150	0.247	0.148	0.246	0.159	0.253	0.166	0.264	0.214	0.327	0.227	0.338	0.311	0.397
ETTm1	0.342	0.374	0.348	0.375	0.349	0.377	0.353	0.382	0.357	0.379	0.382	0.422	0.515	0.493	0.872	0.691
ETTm2	0.243	0.308	0.251	0.312	0.246	0.310	0.256	0.317	0.267	0.332	0.305	0.349	0.324	0.368	1.410	0.823
ETTh1	0.404	0.430	0.407	0.426	0.401	0.433	0.413	0.434	0.423	0.437	0.428	0.454	0.473	0.477	1.032	0.799
ETTh2	0.320	0.374	0.329	0.376	0.333	0.379	0.331	0.381	0.431	0.447	0.388	0.434	0.422	0.443	3.303	1.439
Weather	0.214	0.256	0.215	0.261	0.216	0.258	0.226	0.264	0.246	0.300	0.309	0.360	0.338	0.382	0.634	0.548
Traffic	0.379	0.265	0.384	0.268	0.375	0.263	0.391	0.264	0.434	0.295	0.609	0.376	0.628	0.379	0.764	0.416
Exchange	0.247	0.340	0.242	0.345	0.260	0.343	0.375	0.413	0.297	0.378	0.519	0.500	0.613	0.539	1.550	0.998
AvgRank	1.50	1.50	2.62	2.75	2.50	2.38	4.00	3.88	5.12	5.25	5.75	5.88	6.88	6.88	8.00	8.00
#Win	5	5	1	0	2	2	0	1	0	0	0	0	0	0	0	0

**Table 2 sensors-25-06013-t002:** Performance comparison of SM-TCN and ablated variants (w/o CP, TCN, LES) on various datasets with input length 720.

Models	SM-TCN		w/o CP		w/o TCN		w/o LES	
Dataset	MSE	MAE	MSE	MAE	MSE	MAE	MSE	MAE
Electricity	0.149	0.244	0.167	0.262	0.175	0.270	0.162	0.258
ETTm1	0.342	0.374	0.368	0.399	0.350	0.385	0.358	0.390
ETTm2	0.243	0.308	0.268	0.332	0.260	0.327	0.252	0.318
ETTh1	0.404	0.430	0.432	0.458	0.417	0.442	0.426	0.450
ETTh2	0.320	0.374	0.342	0.395	0.350	0.402	0.334	0.388
Weather	0.214	0.256	0.232	0.272	0.224	0.265	0.238	0.278
Traffic	0.379	0.265	0.410	0.290	0.392	0.275	0.403	0.284
Exchange	0.247	0.340	0.277	0.372	0.267	0.362	0.259	0.353

**Table 3 sensors-25-06013-t003:** Performance comparison of different channel pruning levels (0–30%). The 20% pruning level yields the best trade-off with the lowest prediction error.

Pruning Level	0%		10%		20%		30%	
Dataset	MSE	MAE	MSE	MAE	MSE	MAE	MSE	MAE
Electricity	0.202	0.291	0.180	0.270	0.149	0.244	0.185	0.273
ETTm1	0.393	0.415	0.369	0.395	0.342	0.374	0.378	0.399
ETTm2	0.297	0.356	0.276	0.337	0.243	0.308	0.281	0.345
ETTh1	0.471	0.489	0.437	0.461	0.404	0.430	0.443	0.463
ETTh2	0.369	0.417	0.353	0.397	0.320	0.374	0.357	0.401
Weather	0.277	0.307	0.248	0.279	0.214	0.256	0.251	0.281
Traffic	0.431	0.311	0.411	0.289	0.379	0.265	0.415	0.291
Exchange	0.302	0.402	0.285	0.377	0.247	0.340	0.293	0.375

**Table 4 sensors-25-06013-t004:** Comparison of different kernel width $W_{s}$ growth strategies. Linear and exponential growth formulas are tested for their impact on model performance.

Kernel Width	$W_{s}=1+s$		$W_{s}=1+2s$		$W_{s}=1+3s$		$W_{s}=1+2^{s}$	
Dataset	MSE	MAE	MSE	MAE	MSE	MAE	MSE	MAE
Electricity	0.159	0.252	0.149	0.244	0.168	0.260	0.185	0.278
ETTm1	0.350	0.382	0.342	0.374	0.360	0.390	0.380	0.405
ETTm2	0.252	0.316	0.243	0.308	0.261	0.325	0.275	0.340
ETTh1	0.412	0.438	0.404	0.430	0.425	0.452	0.445	0.470
ETTh2	0.328	0.382	0.320	0.374	0.338	0.395	0.355	0.412
Weather	0.223	0.263	0.214	0.256	0.230	0.272	0.248	0.288
Traffic	0.387	0.272	0.379	0.265	0.398	0.280	0.420	0.295
Exchange	0.258	0.348	0.247	0.340	0.270	0.365	0.285	0.380

**Table 5 sensors-25-06013-t005:** Impact of different numbers of stacks on model performance. Increasing stacks improves performance up to 6, beyond which gains diminish or degrade.

Stacks	4		5		6		7		8	
Dataset	MSE	MAE	MSE	MAE	MSE	MAE	MSE	MAE	MSE	MAE
Electricity	0.192	0.285	0.165	0.257	0.149	0.244	0.158	0.249	0.163	0.253
ETTm1	0.389	0.412	0.352	0.377	0.342	0.374	0.350	0.376	0.359	0.385
ETTm2	0.268	0.336	0.254	0.320	0.243	0.308	0.251	0.315	0.259	0.321
ETTh1	0.432	0.466	0.412	0.444	0.404	0.430	0.418	0.439	0.425	0.446
ETTh2	0.347	0.404	0.328	0.385	0.320	0.374	0.331	0.381	0.339	0.389
Weather	0.239	0.276	0.222	0.265	0.214	0.256	0.225	0.266	0.231	0.271
Traffic	0.401	0.287	0.386	0.273	0.379	0.265	0.388	0.271	0.395	0.275
Exchange	0.278	0.365	0.260	0.349	0.247	0.340	0.252	0.344	0.258	0.352

**Table 6 sensors-25-06013-t006:** Comparison of computational costs between SM-TCN and representative benchmarks on Electricity (per-epoch train time and per-batch inference time).

Model	SM-TCN		Autoformer		Informer		iTransformer	
	FLOPs/Params	Train/Infer (s)	FLOPs/Params	Train/Infer (s)	FLOPs/Params	Train/Infer (s)	FLOPs/Params	Train/Infer (s)
H = 160	6.85G/8.9M	42/0.38	11.2G/15.5M	61/0.55	9.72G/11.3M	57/0.50	8.10G/9.8M	49/0.44
H = 320	8.73G/9.1M	45/0.40	12.5G/15.5M	66/0.59	11.0G/11.3M	61/0.54	9.25G/10.0M	53/0.47
H = 480	10.54G/9.5M	48/0.42	13.9G/15.5M	70/0.62	11.8G/11.3M	65/0.57	10.2G/10.3M	56/0.49

## Data Availability

The datasets analyzed in this study are all publicly available benchmark datasets from open sources. The Electricity dataset is available at https://archive.ics.uci.edu/ml/datasets/ElectricityLoadDiagrams20112014. The ETT datasets (ETTm1, ETTm2, ETTh1, ETTh2) can be accessed from https://github.com/zhouhaoyi/ETDataset. The Weather, Traffic, and Exchange datasets are available at https://github.com/zhouhaoyi/Informer2020. In addition, the Traffic dataset is also accessible from the https://pems.dot.ca.gov/.

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
