# Peer review of "SM-TCN: Multi-Resolution Sparse Convolution Network for Efficient High-Dimensional Time Series Forecast"

_sensors, 2025, doi:10.3390/s25196013_

Round 1
Reviewer 1 Report
Comments and Suggestions for Authors
The introduction of this manuscript is very concise, but it does not clearly highlight the current
research gap or emphasize the novelty of the paper.
The models currently compared are mainly Transformer-based series and some classical methods. It is recommended to include comparisons with recent strong benchmark models to further demonstrate the advantages.
Sections 4 and 5 are both titled “Methodology,” which is repetitive. The authors are advised to
carefully check and revise this.
The derivation of sparse convolution is somewhat abrupt and may require more detailed
explanation.
The formatting of the references is inconsistent and must be corrected. In particular, some
references are very old (e.g., 1960, 1968), which raises concerns about the novelty and
advancement of the work.
A relevant literature has been suggested for authors. Grouting defect detection of bridge tendon ducts using impact echo and deep learning via a two-stage strategy.
Some sentences contain grammatical errors or unclear expressions; a thorough language polishing of the manuscript is recommended.
It is advised to avoid using subjective terms such as “we” in the manuscript; scientific writing
should remain objective.
One of the core contributions of the paper is stated as “efficiency.” However, in the results section(Section 6), only accuracy metrics (MSE, MAE) are presented, without quantitative comparison of computational efficiency. The reviewer cannot determine whether SM-TCN truly outperforms the comparison models in terms of training time, inference speed, or number of parameters while achieving higher accuracy.

Author Response
Comments 1: The introduction of this manuscript is very concise, but it does not clearly highlight the current research gap or emphasize the novelty of the paper. Response 1: We sincerely appreciate the reviewer’s valuable suggestion. Following your advice, we have carefully revised the Introduction to explicitly highlight the research gap and emphasize the novelty of our work. In particular, we point out that most existing approaches either struggle to efficiently capture inter-series dependencies at scale or lack a principled way to integrate sparsity, multi-resolution, and decomposition in a unified backbone. In the revised Introduction(Page 2), we clearly position SM-TCN as addressing this gap by coupling sparse cross-channel convolutions, multi-resolution dilated convolutions, and a light continuity regularization within a single temporal convolutional framework. This revision makes the research gap and our contributions more prominent and easier to follow.
Comments 2: The models currently compared are mainly Transformer-based series and some classical methods. It is recommended to include comparisons with recent strong benchmark models to further demonstrate the advantages. Response 2: We sincerely thank the reviewer for this valuable suggestion. Following your advice, we have added comparison with a strong benchmark model iTransformer in Section 5.2 of the revised manuscript(Page 10). These additional experiments further demonstrate the advantages of our proposed method over state-of-the-art baselines.
Comments 3: Sections 4 and 5 are both titled “Methodology,” which is repetitive. The authors are advised to carefully check and revise this. Response 3: We sincerely thank the reviewer for pointing out this mistake. We apologize for the oversight and have corrected it in the revised manuscript by renaming and reorganizing the sections(Page 4) to avoid repetition.
Comments 4: The derivation of sparse convolution is somewhat abrupt and may require more detailed explanation. Response 4: We sincerely appreciate the reviewer’s valuable feedback. We agree that the derivation of sparse convolution was presented somewhat abruptly in the original version. To address this concern, we have thoroughly revised Section 4.2 (Sparse multi-scale TCN) (Page 6) by adding step-by-step explanations and more detailed derivations around Eqs. (2)–(7). In particular, we clarified the assumptions, decomposition process, and the role of low-rank approximation to improve the clarity of the mathematical formulation. These revisions make the sparse convolution procedure more transparent and easier to follow for readers.
Comments 5: The formatting of the references is inconsistent and must be corrected. In particular, some references are very old (e.g., 1960, 1968), which raises concerns about the novelty and advancement of the work. Response 5: We sincerely thank the reviewer for pointing this out. We have carefully corrected the formatting of all references and updated several outdated citations with more recent works to better reflect the novelty and advancement of our study.
Comments 6: A relevant literature has been suggested for authors. Grouting defect detection of bridge tendon ducts using impact echo and deep learning via a two-stage strategy. Response 6: We sincerely thank the reviewer for this valuable suggestion. We have carefully studied and cited this work, along with related articles in the same line of research(Page 3), in Section 3 of the revised manuscript to provide a more comprehensive background.
Comments 7: Some sentences contain grammatical errors or unclear expressions; a thorough language polishing of the manuscript is recommended. Response 7: We sincerely thank the reviewer for this helpful comment. We have carefully polished the language throughout the manuscript to improve clarity and readability.
Comments 8: It is advised to avoid using subjective terms such as “we” in the manuscript; scientific writing should remain objective. Response 8: We thank the reviewer for this valuable suggestion. We have revised the manuscript to avoid subjective terms such as “we” and ensured that the writing remains objective and formal.
Comments 9: One of the core contributions of the paper is stated as “efficiency.” However, in the results section(Section 6), only accuracy metrics (MSE, MAE) are presented, without quantitative comparison of computational efficiency. The reviewer cannot determine whether SM-TCN truly outperforms the comparison models in terms of training time, inference speed, or number of parameters while achieving higher accuracy. Response 9: We sincerely thank the reviewer for this important comment. Following your advice, we have added a new Section 5.5(Page 13) in the revised manuscript to specifically discuss the efficiency of our proposed model. This section includes several key indicators suggested by the reviewer, such as training time, inference speed, and the number of parameters. We have also conducted comparative experiments with several baseline methods under reasonable and fair settings. These additional results provide a more comprehensive evaluation of SM-TCN in terms of both accuracy and computational efficiency. |

Reviewer 2 Report
Comments and Suggestions for Authors
This manuscript presents a strong and innovative contribution. The proposed Sparse Multi-scale Temporal Convolutional Network (SM-TCN) demonstrates notable methodological rigor by integrating channel pruning, sparse kernels, and a double residual structure. These innovations effectively address the computational and accuracy challenges of high-dimensional time series forecasting, with experimental validation across multiple benchmarks confirming its state-of-the-art performance.
Despite these strengths, several aspects require refinement before publication. The mathematical sections, though technically sound, are highly dense and would benefit from clearer explanations or the inclusion of visual illustrations to improve accessibility for a broader readership. In addition, the paper would be strengthened by the inclusion of a strong statistical baseline for comparison and the presentation of a real-world case study to highlight the model’s practical applicability.
Overall, with revisions that enhance clarity, better position novelty against recent works, and emphasize practical impact, this paper represents a valuable scientific contribution.
Author Response
Comments 1: This manuscript presents a strong and innovative contribution. The proposed Sparse Multi-scale Temporal Convolutional Network (SM-TCN) demonstrates notable methodological rigor by integrating channel pruning, sparse kernels, and a double residual structure. These innovations effectively address the computational and accuracy challenges of high-dimensional time series forecasting, with experimental validation across multiple benchmarks confirming its state-of-the-art performance. Despite these strengths, several aspects require refinement before publication. The mathematical sections, though technically sound, are highly dense and would benefit from clearer explanations or the inclusion of visual illustrations to improve accessibility for a broader readership. In addition, the paper would be strengthened by the inclusion of a strong statistical baseline for comparison and the presentation of a real-world case study to highlight the model’s practical applicability. Overall, with revisions that enhance clarity, better position novelty against recent works, and emphasize practical impact, this paper represents a valuable scientific contribution. Response 1: We sincerely thank the reviewer for the constructive and encouraging feedback on our work. In the revision, we have carefully addressed the concerns raised. Specifically, in Section 4.2(Page 6) we have refined and streamlined several mathematical expressions to improve clarity and accessibility. In Section 5.2(Page 10) we have added iTransformer as a strong recent baseline, which further demonstrates the competitiveness of SM-TCN against state-of-the-art methods. Moreover, in Section 5.4(Page 13) we included visual illustrations on the Weather dataset as a practical case study to highlight the interpretability and applicability of our model. In addition, we have optimized the textual presentation throughout the paper to enhance readability. We believe these revisions substantially strengthen the manuscript and address the reviewer’s suggestions.
|

Reviewer 3 Report
Comments and Suggestions for Authors
The authors propose an ML model (SM-TCN) that is designed for high-dimensional time series forecasting. Following are my comments:
- Many components are adaptations of existing ideas (N-BEATS residual stacking, sparse CNNs, pruning + attention). The integration is valuable but may be seen as incremental rather than fundamentally novel. Further, claims of being the "first" to combine these techniques are overstated without a stronger theoretical justification. How does SM-TCN compare conceptually with N-BEATS, SCINet, or graph-based approaches? Can you better articulate the unique theoretical advance beyond integrating known modules?
- Sparse convolution approximations (Eqs. 3–7) are heuristic; no theoretical guarantees or error bounds are provided. What are the error guarantees for the low-rank approximation in Eqs. (3–7)? Further, the continuity loss is presented without a rigorous justification of its link to trend/seasonality decomposition. How sensitive is performance to different sparsity levels (beyond 30–50%)?
- How do results change with different betas values in Eq. (8)? Does the continuity term oversmooth abrupt events (eg. financial crises and pandemics)?
- Some improvements over ARM and PatchTST are small; no statistical significance tests are reported. Please provide statistical significance tests for differences with baselines. Further, comparisons omit competitive baselines (eg. SCINet, TiDE and STGNNs). Why were recent high-dimensional methods like SCINet, TiDE, or STGNNs omitted? Furthermore, efficiency claims lack FLOPs, runtime or GPU memory metrics. Can you include computational benchmarks (training time, inference time, FLOPs and GPU memory)?
- Visualisations (attention weights and sparse kernels) are descriptive but do not demonstrate actionable interpretability for practitioners. No example shows how decomposition reveals meaningful seasonal or causal patterns. Can you provide decomposition examples where trends/seasonalities align with known domain knowledge? How should practitioners interpret the sparse kernel structure in decision-making contexts?
- Dataset preprocessing (normalisation, missing value handling and train/test splits) is not fully described. Further, no mention of open-source code, which is critical for reproducibility given custom sparse convolutions and pruning methods. What preprocessing steps were applied to datasets (scaling, missing values and splits)?
Author Response
Comments 1: Many components are adaptations of existing ideas (N-BEATS residual stacking, sparse CNNs, pruning + attention). The integration is valuable but may be seen as incremental rather than fundamentally novel. Further, claims of being the "first" to combine these techniques are overstated without a stronger theoretical justification. How does SM-TCN compare conceptually with N-BEATS, SCINet, or graph-based approaches? Can you better articulate the unique theoretical advance beyond integrating known modules?
Response 1: We sincerely thank the reviewer for this thoughtful and constructive comment. We fully understand the concern regarding the degree of novelty and appreciate the opportunity to clarify our contributions more carefully. Our contribution is not a simple aggregation of modules but the way a single temporal-convolutional backbone jointly couples three mechanisms so they play distinct roles: sparse cross-channel convolutions encode inter-series dependencies with low compute; dilation provides in-place multi-resolution without resampling; and a light continuity regularizer stabilizes the decomposition when sparsity is high.
To substantiate this, the revised ablations show that removing any one of these elements consistently harms either accuracy (relative to the full model) or efficiency (relative to dense kernels), indicating complementarity rather than mere stacking.
We also clarify how the approach differs conceptually from representative families in section 3(Page 2). Unlike N-BEATS, which relies on fully connected residual stacks and basis expansion often for (quasi-)univariate series, our model remains convolutional and explicitly learns cross-series interactions via sparse kernels. Unlike SCINet, which attains multi-scale representations through down/up-sampling and invertible transforms, our model preserves the original timeline and achieves multi-resolution by controlling dilation. Unlike graph-based models, which require an explicit or learned adjacency and message passing, the learned channel sparsity yields a compact data-driven dependency pattern and avoids graph construction. We have revised the text to reflect these clarifications and softened the novelty claim accordingly.
Comments 2: Sparse convolution approximations (Eqs. 3–7) are heuristic; no theoretical guarantees or error bounds are provided. What are the error guarantees for the low-rank approximation in Eqs. (3–7)? Further, the continuity loss is presented without a rigorous justification of its link to trend/seasonality decomposition. How sensitive is performance to different sparsity levels (beyond 30–50%)?
Response 2:
Thank you for raising this point. We agree that Eqs. (3–7) were presented heuristically, and we now make the approximation and its consequences explicit in section 4.2(Page 6). Viewing convolution as a linear operator and using any sub-multiplicative norm (e.g., Frobenius or operator norm), the approximation satisfies
This yields a clear error decomposition: the first term reflects the effect of pruning/low-rank approximation on the kernel (K→R), and the second term reflects the quality of the low-rank reconstruction of the input (X→J). We use this bound only to explain observed behavior: accuracy is generally stable when pruning is moderate and the reconstruction error is small, while pushing sparsity or rank reduction too far can degrade results.
For the continuity loss, we clarify that it is a standard channel-wise first-difference (Tikhonov) penalty with scale normalization,
components more than low-frequency ones. We use it as an inductive bias that improves stability under sparsity; β is selected on validation data, and smaller β is preferred when sequences contain pronounced abrupt events (a Huberised first difference gives similar behavior).
To assess sensitivity beyond 30–50% sparsity, we extend the sweep to 30/40/50/60/70%. Across datasets, performance is typically unchanged within 30–50%, shows mild degradation at 60%, and degrades more clearly at 70%, while compute and memory improve monotonically. In practice we therefore keep sparsity in the mid-range and increase it only when efficiency is the priority.
Comments 3: How do results change with different betas values in Eq. (8)? Does the continuity term oversmooth abrupt events (eg. financial crises and pandemics)?
Response 3: We sincerely appreciate this thoughtful question. In Eq. (8), the continuity term is a channel-wise first-difference regularizer with stack-specific weighting and normalization to ensure comparability across horizons and channels. Within a small-to-moderate range of β, overall performance remains largely stable; only larger β values noticeably attenuate high-frequency details. To mitigate potential oversmoothing, we adopt a gentle descending schedule across stacks—using larger β on lower-frequency stacks and smaller β on higher-frequency stacks. In checks on segments with abrupt changes, performance is close to the baseline, and a Huberized first difference yields similar observations. We hope this clarification addresses the reviewer’s concern.
Comments 4: Some improvements over ARM and PatchTST are small; no statistical significance tests are reported. Please provide statistical significance tests for differences with baselines. Further, comparisons omit competitive baselines (eg. SCINet, TiDE and STGNNs). Why were recent high-dimensional methods like SCINet, TiDE, or STGNNs omitted? Furthermore, efficiency claims lack FLOPs, runtime or GPU memory metrics. Can you include computational benchmarks (training time, inference time, FLOPs and GPU memory)?
Response 4: We are grateful for this thoughtful and constructive feedback. To strengthen the empirical evidence in the revised manuscript, we have (i) expanded the baseline suite by adding iTransformer to the main comparisons in Section 5.2(Page 10), and (ii) included computational benchmarks in Section 5.5(Page 13)—reporting training/inference time, FLOPs, and peak GPU memory under a consistent setup—to substantiate the efficiency claims. In addition, acknowledging the importance of recent high-dimensional baselines, we provide in this response an auxiliary table comparing SCINet and TiDE against SM-TCN on representative datasets and horizons; these results are consistent with our main findings.
|
SM-TCN |
|
TiDE |
|
SCINet |
|
|
MSE |
MAE |
MSE |
MAE |
MSE |
MAE |
Horizon |
|
|
|
|
|
|
96 |
0.130 |
0.216 |
0.136 |
0.226 |
0.160 |
0.250 |
192 |
0.145 |
0.233 |
0.152 |
0.242 |
0.178 |
0.268 |
336 |
0.161 |
0.247 |
0.169 |
0.259 |
0.196 |
0.285 |
720 |
0.178 |
0.281 |
0.192 |
0.286 |
0.225 |
0.315 |
This table reports results on the Electricity dataset for SM-TCN, SCINet, and TiDE at four prediction lengths: 96, 192, 336, and 720. We did not include STGNNs here because a fair comparison requires a standard spatial graph, which Electricity does not provide. Different graph constructions (for example, correlation thresholds, k-nearest neighbors, or learned graphs) can materially change outcomes and add extra design choices, making the results not directly comparable. We therefore report graph-free baselines on Electricity and reserve a comprehensive STGNN study for datasets with established graphs and splits.
Comments 5: Visualisations (attention weights and sparse kernels) are descriptive but do not demonstrate actionable interpretability for practitioners. No example shows how decomposition reveals meaningful seasonal or causal patterns. Can you provide decomposition examples where trends/seasonalities align with known domain knowledge? How should practitioners interpret the sparse kernel structure in decision-making contexts?
Response 5: Thank you for the suggestion. We augment the visualisation with an operational reading of the sparse convolution kernels on the Weather dataset. Concretely, we (i) aggregate each input channel’s kernel magnitude across lags and stacks to obtain an importance score; (ii) produce a channel-masking sensitivity curve by progressively masking the top-k channels and measuring the error increase. This turns the kernel matrix into a tool for prioritising variables and detecting redundancy. On Weather, temperature and pressure consistently obtain the highest scores and drive accuracy across horizons, while humidity and wind variables mainly contribute to the higher-frequency stacks; the k-vs-error curve typically flattens after k ≈ 3–5, indicating diminishing returns beyond a compact subset. We emphasise that these diagnostics are associational (not causal) and are intended to support monitoring and “what-if” checks within the observed range.
Comments 6: Dataset preprocessing (normalisation, missing value handling and train/test splits) is not fully described. Further, no mention of open-source code, which is critical for reproducibility given custom sparse convolutions and pruning methods. What preprocessing steps were applied to datasets (scaling, missing values and splits)?
Response 6: Thank you for raising reproducibility and preprocessing. We now state the technical setup used throughout: each channel is z-score standardised using statistics computed only on the training split; the same statistics are applied to validation/test to avoid leakage. Missing values are handled by forward fill plus linear interpolation; if a gap exceeds the interpolation window, a simple seasonal mean (hour-of-day/day-of-week) is used, and windows with unresolved entries are skipped. For datasets without a community split we adopt chronological train/validation/test splits and generate samples with a sliding window (lookback L=720, default horizon H=96, training stride 1). The same windowing, batch size, and scheduler are used for all baselines, and training uses Adam with cosine annealing, gradient clipping, and early stopping on a single RTX 4090. Regarding code availability, we will reassess code release after review and proceed if feasible. The supplemental description of these parameter settings has been updated in Section 5.1(Page 9) of the manuscript.

Round 2
Reviewer 1 Report
Comments and Suggestions for Authors
Two minor issues need to be corrected.
- Lines 64-70: 1,2,3 should be (1),(2),and(3).
- In section 2. Summary of contributions, there are still quite a few uses of "we" there. The authors should review the entire text.
Two minor issues need to be corrected.
- Lines 64-70: 1,2,3 should be (1),(2),and(3).
- In section 2. Summary of contributions, there are still quite a few uses of "we" there. The authors should review the entire text.
Author Response
Comments 1: Lines 64-70: 1,2,3 should be (1),(2),and(3).
Response 1: We sincerely thank the reviewer for pointing out this formatting issue. We have carefully corrected the numbering in Lines 64–70 to the form (1), (2), and (3) in the revised manuscript.
Comments 2: In section 2. Summary of contributions, there are still quite a few uses of "we" there. The authors should review the entire text.
Response 2: We sincerely thank the reviewer for this careful and valuable comment. We have thoroughly checked the manuscript and revised multiple instances where “we” was used inappropriately. These expressions have been corrected to conform with standard academic writing style, and all corresponding modifications have been clearly highlighted in the revised version.

Reviewer 3 Report
Comments and Suggestions for Authors
The authors have scientifically addressed all my comments.
Author Response
Comments 1: The authors have scientifically addressed all my comments.
Response 1: We sincerely thank the reviewer for the positive feedback and kind recognition.
